# Evaluation of Intraocular Pressure and Other Biomechanical Parameters to Distinguish between Subclinical Keratoconus and Healthy Corneas

**DOI:** 10.3390/jcm10091905

**Published:** 2021-04-28

**Authors:** Cristina Peris-Martínez, María Amparo Díez-Ajenjo, María Carmen García-Domene, María Dolores Pinazo-Durán, María José Luque-Cobija, María Ángeles del Buey-Sayas, Susana Ortí-Navarro

**Affiliations:** 1FISABIO Oftalmología Médica (FOM), Anterior Segment and Cornea and External Eye Diseases Unit, Bifurcación Pío Baroja-General Avilés, 12, E-46015 Valencia, Spain; cristinaperismartinez0@gmail.com (C.P.-M.); amparo.diez@uv.es (M.A.D.-A.); m.carmen.garcia-domene@uv.es (M.C.G.-D.); maria.j.luque@uv.es (M.J.L.-C.); 2Surgery Department, Ophthalmology, School of Medicine, University of Valencia, Av. Blasco Ibáñez, 15, E-46010 Valencia, Spain; pinazoduran@yahoo.es; 3Aviño Peris Eye Clinic, Avinguda de l’Oest, 34, E-46001 Valencia, Spain; 4Optics, Optometry and Vision Sciences Department, School of Physics, University of Valencia, Dr. Moliner, 50, E-46100 Valencia, Spain; 5Hospital Lozano Blesa, Anterior Segment and Cornea and External Eye Diseases, E-46015 Zaragoza, Spain; madelbuey@gmail.com

**Keywords:** intraocular pressure, ocular inflammation, cornea biomechanics, Corvis^®^ ST, subclinical keratoconus

## Abstract

(1) Purpose: To assess the main corneal response differences between normal and subclinical keratoconus (SCKC) with a Corvis^®^ ST device. (2) Material and Methods: We selected 183 eyes of normal patients, of a mean age of 33 ± 9 years and 16 eyes of patients with SCKC of a similar mean age. We measured best corrected visual acuity (BCVA) and corneal topography with a Pentacam HD device to select the SCKC group. Biomechanical measurements were performed using the Corvis^®^ ST device. We carried out a non-parametric analysis of the data with SPSS software (Wilcoxon signed rank-test). (3) Results: We found statistically significant differences between the control and SCKC groups in some corneal biomechanical parameters: first and second applanation time (*p* = 0.05 and *p* = 0.02), maximum deformation amplitude (*p* = 0.016), highest concavity radius (*p* = 0.007), and second applanation length and corneal velocity ((*p* = 0.039 and *p* = 0.016). (4) Conclusions: Our results show that the use of normalised biomechanical parameters provided by noncontact tonometry, combined with a discriminant function theory, is a useful tool for detecting subclinical keratoconus.

## 1. Introduction

Knowledge of corneal biomechanics is essential to understand corneal behaviour in certain diseases, surgical procedures, intraocular pressure (IOP) measurements, and in the early detection and treatment of subclinical keratoconus (SCKC).

Keratoconus is a bilateral, inflammatory, asymmetric and progressive corneal ectasia disorder. Bowman’s layer in keratoconus patients is impaired, associated changes in the stromal extracellular matrix are brought about [1], and a cycle of thinning and increased strain occurs [2,3]. The collagen network is mostly unorganised, with decreased fibrillar interweaving [3,4,5]. These changes reduce corneal stiffness [2,5].

The most commonly used device for analysing corneal biomechanical parameters is the Ocular Response Analyzer (ORA^®^, Reichert Ophthalmic Instruments, Inc., Buffalo, NY, USA) [4,5,6,7,8]. Some studies found a good correlation between keratoconus and low corneal hysteresis (CH) and corneal resistance factor (CRF) in high grade keratoconus [7,8,9,10,11].

The Corvis^®^ ST device (Oculus Optikgeräte GmbH; Wetzlar, Germany) is a non-contact tonometer system with an ultra-high-speed Scheimpflug camera that provides corneal biomechanical parameters and IOP information [12,13,14,15,16,17,18,19,20,21,22,23,24,25,26]. It also affords a high degree of repeatability [13,25,26,27] and gives a correlation between age, corneal thickness, IOP, and some Corvis^®^ ST biomechanical parameters [17,24,28].

Some studies show differences in biomechanical Corvis^®^ ST parameters between keratoconic and healthy corneas [12,13,14,15,16,17,18,19,20,21,22,23]. More recently, other studies have evaluated SCKC with this device [12,18,19,29,30].

Early SCKC detection is important since treatment with collagen cross-linking can slow the progression of keratoconus effectively. If subtle biomechanical changes in early keratoconus go undetected, advanced keratoconus treatment could be delayed. In addition, proper patient selection (without SCKC) is essential for the success of refractive surgery. Clearly, it is difficult to distinguish between SCKC and eyes with healthy corneas when only using slit-lamp criteria. Topographic values only provide information of static changes and it must be taken into account that air pressure–corneal deformation is affected in keratoconus patients [7,12,15,31].

The main purpose of the present study was to identify the most useful parameters provided by non-contact tonometry for the biomechanical characterization of the cornea and to determine whether it is possible to define an optimized function for SCKC detection.

## 2. Materials and Methods

This study adheres to the tenets of the Declaration of Helsinki for Research Involving Human Subjects and was approved by our Institutional Review Board. This retrospective, consecutive, non-randomised study analyses 199 eyes of 196 patients using the Corvis^®^ ST tonometer. The eyes were divided into two groups: (a) healthy eyes (183 eyes of 183 patients); and (b) eyes with subclinical keratoconus (16 eyes of 13 patients). The eyes with subclinical keratoconus fulfilled the most widely accepted definition in the literature for this condition [30,32,33]. These eyes had no clinical signs of keratoconus (Vogt’s striae, Fleischer rings or corneal scarring), their topography was normal with no asymmetric bowtie, and no focal or inferior steepening pattern. However, they were contralateral eyes of clinically evident keratoconus in the fellow eye [32]. Three of them were considered as bilateral SCKC.

The inclusion criteria for both groups were the non-use of contact lenses during the previous four weeks if such contact lenses were rigid, or two weeks if they were soft, and ages between 18 and 50.

Exclusion criteria were previous ophthalmic surgery, any ocular or systemic disease, corneal scars and/or opacities, chronic use of topical medication, pregnancy or refusal to sign the informed consent form.

We measured the corneal topography of all the patients with a Pentacam HD device (Oculus, Wetzlar, Germany). The corneal status was established by slit-lamp microscopy and analysed and classified by an experienced ophthalmologist. The control patients had no clinical keratoconus symptoms and their corneal topography was within normal limits. Diagnosis of SCKC was made when eyes had no clinical signs of keratoconus (Vogt’s striae, Fleischer rings or corneal scarring), their topography was normal with no asymmetric bowtie (with a paracentral inferior–superior dioptric difference less than 1 D), and no focal or inferior steepening pattern. We included patients with the steepest meridian under 47.2 D who did not present clinical signs [30,32,33].

Best corrected visual acuity (BCVA) was measured with an ETDRS chart. The Corvis^®^ ST evaluated corneal biomechanics. This device can identify the applanation time and length and corneal applanation velocity when the air pulse is on (A1time, A1length and A1V, respectively) and off (A2time, A2length and A2V, respectively). It can also identify the highest concavity time (HCtime), the deformation amplitude (DA_max_), the peak distance (PD) and the curvature radius (R_HC_) at the highest concavity (HC). All these data are obtained from the dynamic corneal deformation during a defined air pulse. Central corneal thickness (CCT) is also calculated using the horizontal Scheimpflug image at the apex. Intraocular pressure is calculated based on the timing of the first applanation event. The Corvis biomechanical index (CBI) was not evaluated because the updated version was not available on our Corvis^®^ ST device when the measurements were made.

Statistical analysis was performed using the SPSS 26.0 software for Windows (SPSS, Chicago, IL, USA) and principal component analysis (PCA) was carried out with Matlab 2020a (The Mathworks, Inc., Natick, MA, USA). For each variable, values came from the mean of three measurements. The Kolmogorov–Smirnov test was used to check for sample normality. Distributions for the SCKC group failed the normality test, and therefore a non-parametric Wilcoxon signed rank-test was used to compare parameters between the groups. The level of significance used was *p* < 0.05.

## 3. Results

Table 1 summarizes the demographic data of the patients. There was no significant difference in age between the groups.

The control and SCKC group had a BCVA of 0.12 ± 0.20 logMAR and 0.04 ± 0.20 logMAR, respectively (*p* = 0.002). Mean corneal keratometry were 43.0 ± 1.7 D and 44.2 ± 1.9 D for the flattest meridian, in the control and SCKC group, respectively (*p* = 0.328). The steepest meridian was 43.9 ± 1.8 D and 45.0 ± 3.0 D in the control and SCKC group, respectively (*p* = 0.006).

Figure 1a, Figure 2a, Figure 3a and Figure 4a show the values obtained for the first and second applanation, and for the HC. CCT and IOP values were significantly different in the two population samples (*p* < 0.05 for both parameters, with 9.6 ± 2.7 mm Hg and 510 ± 49 μm for the SCKC group and 12.3 ± 2.9 mm Hg and 541 ± 38 μm for the control group). To discount a possible effect of this difference, in Figure 1b, Figure 2b, Figure 3b and Figure 4b, a new control group (n = 53) was defined, matching the CCT and IOP values of the SCKC subjects (517 ± 18 μm and 10.4 ± 1.5 mm Hg). Differences in IOP and CCT between the SCKC and the IOP-CCT matched normals were not significant (*p* = 0.57 and *p* = 0.32 for CCT and IOP, respectively).

The whole control group reached the first applanation significantly later than the SCKC group (*p* = 0.001, see Figure 1a), A1length was greater than the SCKC group (*p* = 0.66, see Figure 2a), and A1V was slower in the control group than in the SCKC group (*p* = 0.24, see Figure 3a), but these changes were not significant.

At HC, DA_max_ (Figure 2a) and the PD (Figure 4a) were smaller in the whole control group than in the SCKC group (*p* = 0.0005 and 0.15, respectively) and R_HC_ (Figure 4) was smaller in the SCKC group than in the whole control group (*p* < 0.001). The HC time was similar in both groups (*p* = 0.85, Figure 1a).

In the second applanation, A2time (Figure 1a) and A2V (Figure 3a) were significantly higher in the SCKC group than in the total control group (*p* = 0.02 and 0.001, respectively), but A2length (Figure 2a) was significantly smaller in the SCKC group than in the control group (*p* = 0.01).

When comparing SCKC with the IOP-CCT matched normals, similar trends are found, but with changes in the significance of the differences. DA_max_ (*p* = 0.006, Figure 2b), A2length (*p* = 0.03, Figure 2b), A2V (*p* = 0.006, Figure 3b), and R_HC_ (*p* = 0.05, Figure 4b) maintained their statistical differences, and differences were also found in A1length (*p* = 0.02, Figure 2b). However, statistical differences were not found in A1time (*p* = 0.34) or A2time (*p* = 0.57) (Figure 1b).

Receiver operating characteristic (ROC) curves were derived, using both the whole normal sample and the IOP and CCT matched normal subjects with the four parameters that yielded significant differences between matched normal and the SCKC group (R_HC_, A2length, A2V, and DA_max_). Figure 5 shows the results obtained.

The best result, according to the area under the curve (AUC), is yielded by A2V, but the best sensitivity to specificity ratio is only 25–75%, approximately. To determine whether a combination of these four parameters would improve these results, we performed a principal component analysis (PCA). None of the four principal components achieved a total separation of the normal and SCKC samples, but the first component, explaining 65% of the variance when the total normal sample is used and 59% with the reduced normal sample, yielded a ROC curve that slightly improves the result of the individual variables (see Table 2). The improvement is noticeable only in the specificity at high sensitivity (80% or higher).

## 4. Discussion

Detection of corneal ectatic disorders as early on as possible is necessary to prevent, or delay, the progression of keratoconus. Some studies claim to distinguish SCKC from normal eyes using topographic parameters [34,35,36] and others state that there is an overlap between topographic data obtained from a SCKC group and a normal group [37,38]. As a result, biomechanical data could improve the detection and severity prediction of keratoconus.

There are studies on corneal biomechanics using the ORA device that conclude that the four parameters measured with this device are not enough to detect keratoconus [21,39]. Alternative ORA parameters related to the area under the curve are better for distinguishing SCKC from normal corneas, and when all the parameters are combined, accuracy increases [34,39]. The ORA device has limitations, because it can confuse corneal tissue response with surface response since a specular reflection is required to measure applanation pressure. Central corneal surface irregularities could interfere with the infrared specular reflection beam of the ORA. The Corvis^®^ ST device avoids some of these drawbacks, because a frontal view camera is mounted with a keratometric-type projection system for focusing and aligning the corneal apex to be measured. Moreover, the recording of corneal deformation prevents the specular reflection beam from obtaining reliable corneal response parameters.

If one simply addresses statistical data, our data are consistent with biomechanical corneal properties. The A1time is shorter in the SCKC group than in the control group, and DA_max_ is greater in the SCKC group than in the control group. This agrees with previous studies, in which very similar values of DA_max_ were obtained in the SCKC and control groups [15,16,30,40]. During corneal recovery, A2time and A2V were higher in the SCKC group, and R_HC_ and A2length were higher in controls, in agreement with several studies in the literature [30,40]. Except for A2length, these parameters could be correlated with a decrease in the viscoelastic structure, and abnormal elastic distensibility increased in keratoconic corneas, which is consistent with reduced corneal stiffness [4,41].

Due to the decrease in corneal stiffness, a shorter A1time, a higher DA_max_, and a lower R_HC_ could be expected in the SCKC group, as well as a longer A2time and A2V, as keratokonic corneas recover more slowly than normal ones, due to higher initial deformation. Although the differences are not significant, the higher PD, A2time, and A1V values in the SCKC group also confirm this hypothesis.

In principle, we could expect greater A2length values in the SCKC group than in the control group as a result of reduced corneal stiffness. It is possible that increased keratoconus cornea distensibility may produce a non-perfect applanation surface, and Corvis^®^ ST only detects a small portion of plane surface. Therefore, it must be taken into account that our study yielded similar results to those obtained in previous studies [16,30,40].

Although we selected patients without clinical signs, BCVA was statistically worse in the SCKC group. This loss of visual acuity may have been due to the wavefront aberrations that can distort visual quality, even at the beginning of the pathology [42].

When the effect of the IOP and CCT parameters are eliminated, Corvis^®^ ST results change. There are parameters that demonstrate their independence of IOP and CCT measurements, such as DA_max_, A2V, and R_HC_, so they can be considered as robust parameters. These three parameters can identify SCKC correctly, but parameters determining time to deformation and recovery are greatly dependent on IOP and CCT. This is no surprise, since it is known that higher CCT corneas have more resistance to deformation, and the same result could be expected with higher IOP eyes. In addition, we obtained significant differences in length applanation values by eliminating CCT and IOP effects. This confirms that these parameters also have a great dependence on IOP and CCT values, since higher values in these parameters could lead to smaller applanation length due to the resistance to deformation that the cornea can present.

When we analyse ROC curves with the four parameters that yielded significant differences with the SCKC group, that is, R_HC_, A2length, A2V and DA_max_, we can conclude that A2V was the best parameter to diagnose SCKC patients. In this small sample, however, a combination of these four parameters, found by principal component analysis, though improving the AUC under the ROC curve would only improve specificity at high sensitivity.

This study has some limitations. Our SCKC sample was small, and we are aware that a greater number of patients with SCKC would be necessary to obtain more reliable values. To calculate ROC curves with a small SCKC sample could lead to inexact conclusions about the best parameters to detect it. However, a similar number of SCKC eyes (between 12 and 23) were evaluated in previously published studies [12,18,21,22,30], because it is difficult to obtain this sample.

## 5. Conclusions

We were able to detect biomechanical impairment in SCKC eyes in clinical examinations by using Corvis^®^ ST parameters. Some of them have demonstrated a great dependence on IOP and CCT, so to make a correct diagnosis of these patients, only parameters without IOP and CCT dependence should probably be compared. However, further measurements in SCKC patients are necessary, and the effect of IOP and CCT must be studied in more detail, to confirm the findings of our study and to improve current SCKC screening.

## Figures and Tables

**Figure 1 jcm-10-01905-f001:**
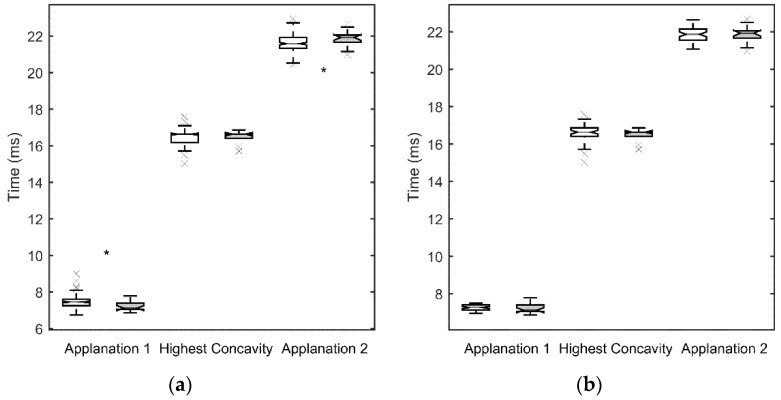
Time to reach first applanation, highest concavity and second applanation for the SCKC (grey) and control groups (white). The middle line of each box is the median of the distribution, the extremes are the first and third quartiles and the whiskers represent 1.5 times the interquartile distance; outliers are plotted with the symbol “x”. The notch represents de 95% confidence interval of the mean. “*” indicates statistically significant differences between groups (*p* < 0.05) (**a**) whole control group, (**b**) IOP and CCT matched control group.

**Figure 2 jcm-10-01905-f002:**
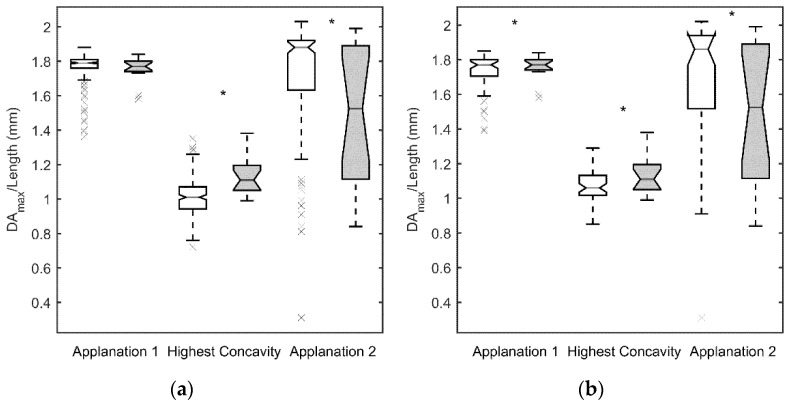
Maximum deformation amplitude (DA_max_) at highest concavity and length applanation at first and second measurement for the SCKC (grey) and control groups (white). Symbols, as in Figure 1. (**a**) whole control group, (**b**) IOP and CCT matched control group.

**Figure 3 jcm-10-01905-f003:**
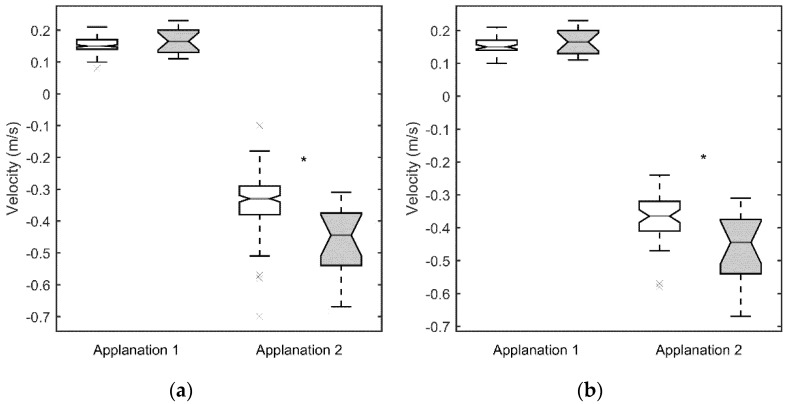
Velocity of deformation at first and second applanation for the SCKC (grey) and control groups (white). Symbols as in Figure 1. (**a**) whole control group, (**b**) IOP and CCT matched control group.

**Figure 4 jcm-10-01905-f004:**
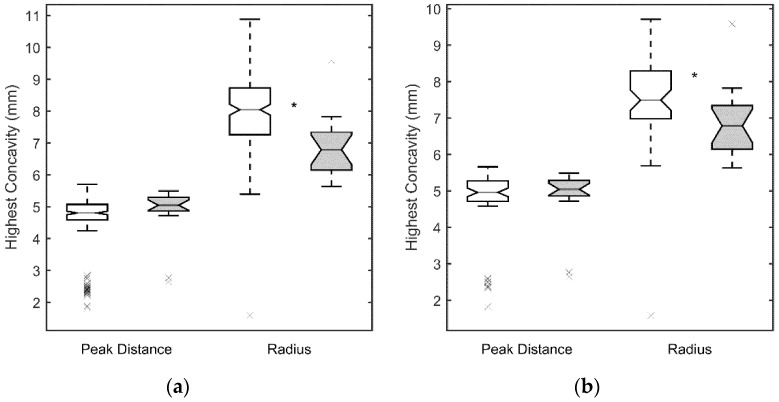
Peak Distance (PD) and radius at highest concavity (R_HC_) for the SCKC (grey) and control groups (white). Symbols as in Figure 1. (**a**) whole control group, (**b**) IOP and CCT matched control group.

**Figure 5 jcm-10-01905-f005:**
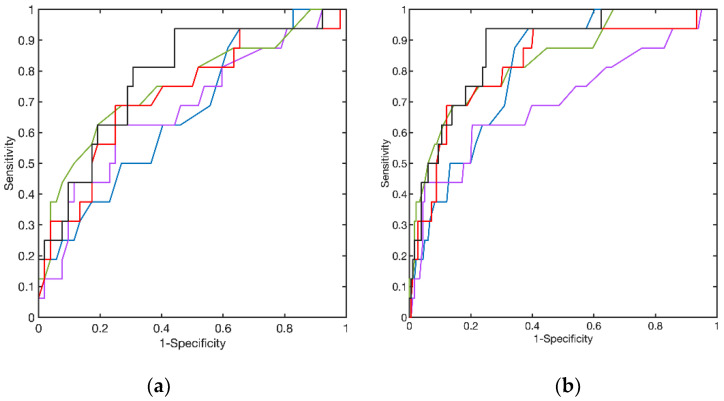
Receiver Operating Characteristic (ROC) curves for High Concavity Radius (red), second applanation length (purple), velocity (green) and maximum amplitude (blue). In black, the curve obtained with the first principal component score of the principal component analysis carried out with these four variables. The dashed line represents the x = y reference line. (**a**) curves obtained with the whole normal sample. (**b**) curves obtained with the IOP and CCP matched normal sample.

**Table 1 jcm-10-01905-t001:** Demographic data of our sample. Age differences between control and SCKC groups were not significant (*p* = 0.55).

Data	Sex	Control Group	SCKC Group
Age (years)	Men	31 ± 7	26 ± 13
	Women	33 ± 8	31 ± 19
*p*-value		0.67	0.50
Number of eyes	Men	83	8
	Women	100	5

**Table 2 jcm-10-01905-t002:** Area under de ROC curve (AUC) and upper and lower limits of the AUC 95% confidence interval for the different diagnosis parameters computed both with the entire normal sample and the IOP and CCP matched normal sample. DA_max_, maximum displacement amplitude, second applanation length (A2length) and corneal velocity (A2V), radius at high concavity (Radius) and the first principal component scores of the principal component analysis (1st PCS). ‘*’ marks AUCs significantly different from 0.5 (no discrimination).

	Total Normal Sample			IOP and CCP Matched Normal Sample		
Variable	AUC	Lower ICL	Upper ICL	AUC	Lower ICL	Upper ICL
1st PCS	0.8695 *	0.7452	0.9269	0.7800 *	0.6073	0.8906
A2V	0.8343 *	0.6948	0.9176	0.7524 *	0.5815	0.8814
Radius	0.8203 *	0.6572	0.9142	0.7230 *	0.5662	0.8506
DA_max_	0.8047 *	0.7052	0.8869	0.6629	0.4710	0.7929
A2length	0.6987 *	0.5244	0.8496	0.6767	0.4970	0.8266

## Data Availability

Data available on request due to restrictions of privacy.

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
