# Peer review of "Evaluation of Intraocular Pressure and Other Biomechanical Parameters to Distinguish between Subclinical Keratoconus and Healthy Corneas"

_jcm, 2021, doi:10.3390/jcm10091905_

Round 1

Reviewer 1 Report

I read with great interest the manuscript titled: ‘’EVALUATION OF INTRAOCULAR PRESSURE AND OTHER BIOMECHANICAL PARAMETERS TO DISTINGUISH BETWEEN SUBCLINICAL KERATOCONUS AND HEALTHY CORNEAS’’

Where authors try to find a way to observe the corneal response differences between normal and subclinical keratoconus (SCKC) with a Corvis® ST device. Authors found biomechanical parameters could diagnose SCKC.

It is an interesting research. This article did not add anything new to the literature. Moreover, the sample of SCKC patients are very small. However, and following other research, the outcomes described here, reinforce those before which were published and highlight that study the biomechanics is important for our patients.

Some comments (mayor): English have to be improved. There are many grammatical errors

Abstract:

-‘’We found statistical differences between’’. I suggest to include the word ‘’significant’’ before statistical

Introduction:

-Although I agree with the authors that KC is an inflammatory disease, I do not know that KC cause an inflammatory degeneration. Please try to change this phrase.

- IOP: need to add intraocular pressure

- CCT: same

-Line 72: when you use at first time SCKC, please continue

Methods:

Authors described SCKC as ´´The eyes with 78 subclinical keratoconus fulfilled the most widely-accepted definition in the literature for 79 this condition32,33. These eyes had no clinical signs of keratoconus (Vogt's striae, Fleischer 80 rings or corneal scarring), their topography was normal with no asymmetric bowtie and 81 no focal or inferior steepening pattern; however, they were contralateral eyes of clinically 82 evident keratoconus in the fellow eye.’’ ‘’

but then used

‘’Diagnosis of SCKC patients was made when corneal topography showed an asym- 94 metric bowtie pattern, with or without skewed axes, and a paracentral inferior–superior 95 dioptric difference above 1D ‘’

For me it makes no sense

Results
Please authors need to add the ‘’p value’’ in the table

For me the bars are quite similar although authors found a difference statistically significant

Maybe a ROC curve could be interesting to show the AUC

Discussion:

When the effect of the IOP and ACT parameters are eliminated. What is ACT?

Author Response

Dear Editor and Reviewers,

Thank you for all your comments, which have materially helped us to improve the manuscript. Here you can find our answers to all your comments, point by point:

Reviewer 1:

  1. Some comments (mayor): English have to be improved. There are many grammatical errors

We have rechecked and largely rewritten the manuscript in order to correct grammatical, typographical and formatting errors. Changes have been highlighted in the manuscript using the "Track Changes" function.

  1. Abstract: ‘’We found statistical differences between’’. I suggest to include the word ‘’significant’’ before statistical

We have introduced this change in the manuscript.

  1. Introduction: Although I agree with the authors that KC is an inflammatory disease, I do not know that KC cause an inflammatory degeneration. Please try to change this phrase.

As reviewer suggested, we changed our original sentence “Keratoconus is a progressive corneal ectasia disorder with bilateral, inflammatory, and asymmetric degeneration.” by “Keratoconus is a bilateral, inflammatory, asymmetric and progressive corneal ectasia disorder.

  1. IOP: need to add intraocular pressure

IOP acronym was defined in the first paragraph of the Introduction.

  1. CCT: same

We have defined this acronym as ‘central corneal thickness’ in the last paragraph of the Introduction.

  1. Line 72: when you use at first time SCKC, please continue

We defined this acronym at the abstract section as ‘subclinical keratoconus’ and in the first paragraph of the Introduction.

  1. Methods: Authors described SCKC as ´´The eyes with 78 subclinical keratoconus fulfilled the most widely-accepted definition in the literature for 79 this condition32,33. These eyes had no clinical signs of keratoconus (Vogt's striae, Fleischer 80 rings or corneal scarring), their topography was normal with no asymmetric bowtie and 81 no focal or inferior steepening pattern; however, they were contralateral eyes of clinically 82 evident keratoconus in the fellow eye.’’ but then used ‘’Diagnosis of SCKC patients was made when corneal topography showed an asym- 94 metric bowtie pattern, with or without skewed axes, and a paracentral inferior–superior 95 dioptric difference above 1D ‘’

For me it makes no sense

Indeed, these two definitions are contradictory. There is a mistake in both definitions, related to how the patients have been selected. Both sentences have been rewritten in our manuscript to clarify this aspect and to rectify our mistake.

  1. Results: Please authors need to add the ‘’p value’’ in the table

We have included in Table 1 the p-values for age differences between men and women in each sample, and in the table legend we have included a comment about age differences between the control and SCKC groups, with its corresponding p -value.

  1. For me the bars are quite similar although authors found a difference statistically significant. Maybe a ROC curve could be interesting to show the AUC

It is difficult to see small but significant differences in the bar plots used. We have replaced the figures by boxplots, with notchs representing the 95% confidence interval of the median. This correlates better with the results of the Kruskal-Wallis test: if the notchs do not coincide, the medians significantly differ. To facilitate comparisons, the plots for the comparison between normals and SCKC y IOP-CCT matched normals and SCKC have been grouped.

We have obtained the ROC curves for those parameters showing statistically significant differences between the normal and SCKC samples. Areas under the curve range from 0.7 and 0.83 and the best sensitivity-specificity values are around 80%-20%, respectively. Principal component analysis has also been performed. The first principal component, explaining 65% of the variance when the whole normal sample is considered and 58% when only IOP and CCP matched controls are used, slightly improves AUC, without modifying the best sensitivity-specificity ratio.

  1. Discussion: When the effect of the IOP and ACT parameters are eliminated. What is ACT?

This acronym is a mistake. The correct acronym is ‘CCT’, central corneal thickness. We replaced it in the text.

Reviewer 2 Report

In this manuscript, Peris-Martínez et al. showed that various corneal biomechanical parameters from Corvis ST device showed significant differences between subclinical keratoconus and healthy corneas, indicating their possible clinical usage as diagnostic tool for detecting subclinical keratoconus. The results from this manuscript may be of readers' interest since authors introduced feasibility of applying biomechanical parameters for diagnostic tool for subclinical keratoconus, but there are major issues to address and revise. 

  1. Line 80-83, authors have indicated definition of subclinical keratoconus included in the study, and they have stated the definition again in line 94-97. Which one is correct? Please specify and included references for the definition.
  2. Line 99-104, the sentence seems too long. Please rephrase the sentence.
  3. In Results, why did authors re-analyzed the data with subgroups after controlling (?) CCT and IOP? If authors wanted to control these parameters, they should have matched these parameters before analyzing at first place.
  4. While authors indicated that A1time and A2time did not show significant differences in subgroup analyses after matching for CCT and IOP, figure 1 and 5 seems same to me. Please explain.

Author Response

Dear Editor and Reviewers,

Thank you for all your comments, which have materially helped us to improve the manuscript. Here you can find our answers to all your comments, point by point:

Reviewer 2:

  1. Line 80-83, authors have indicated definition of subclinical keratoconus included in the study, and they have stated the definition again in line 94-97. Which one is correct? Please specify and included references for the definition.

Indeed, these two definitions of subclinical keratoconus are contradictory. There is a mistake in both definitions, related to how the patients of this study have been selected. Both sentences have been rewritten in our manuscript to clarify this aspect and to rectify our mistake.

  1. Line 99-104, the sentence seems too long. Please rephrase the sentence.

As the reviewer suggested, we have tried to simplify this sentence in the text, which now reads as follow:

This device can identify the applanation time and length and corneal applanation veloc-ity when the air pulse is on (A1time, A1length and A1V, respectively) and off (A2time, A2length and A2V, respectively). It can also identify the highest concavity time (HCtime), the deformation amplitude (DAmax), the peak distance (PD) and the curvature radius (RHC) at the highest concavity (HC). All these data are obtained from the dynamic corneal de-formation during a defined air pulse.

  1. In Results, why did authors re-analyzed the data with subgroups after controlling (?) CCT and IOP? If authors wanted to control these parameters, they should have matched these parameters before analyzing at first place.

At first, our aim was not to control intraocular pressure (IOP) and CCT (central corneal thickness) parameters. We were going to work with subclinical keratoconus (SCKC), with no relevant clinical symptoms different from a healthy population. Moreover, we did not know how many SCKC we were going to find and their characteristics. So, we collected a large control group database. However, it has been shown that IOP and CCT can affect the cornea's biomechanical responses (Kling, Marcos IOVS 2013; Vantipalli et al, 2018). This should be taken into account in the analysis of the CorvisST parameters. The number of SCKC subjects and the large number of variables precluded the use of a generalized linear model to account for the effect of IOP and CCT in the other variables. We found that the differences between normals and SCKC in certain variables dissapear when the normals are matched in CCT and IOP, whereas in other variables the differences remain significant. This second group of variables would be less liable to error due to IOP and CCT differences and might provide more reliable diagnosis of subclinical keratoconus.

Kling S, Marcos S. Contributing factors to corneal deformation in air puff measurements. Invest Ophthalmol Vis Sci. 2013;54(7):5078-85. doi: 10.1167/iovs.13-12509.

Vantipalli S, Li J, Singh M, Aglyamov SR, Larin KV, Twa MD. Effects of Thickness on Corneal Biomechanical Properties Using Optical Coherence Elastography. Optom Vis Sci. 2018;95(4):299-308. doi: 10.1097/OPX.0000000000001193.

  1. While authors indicated that A1time and A2time did not show significant differences in subgroup analyses after matching for CCT and IOP, figure 1 and 5 seems same to me. Please explain.

Mean values and standard differences computed with the two normal samples are too small for figures 1 and 5 to look very different. However, differences between normals and SCKC, though small, are significant when the whole normal sample is used, and not significant when only the IOP-CCT matched normals are considered. We have replaced the figures by boxplots, with notchs representing the 95% confidence interval of the median. Visually, they correlate better with the results of the Kruskal-Wallis test. The differences between the old figures 1 and 5 can be seen in the change in the relative position of the “notches” defining the 95% confidence interval of the median. To facilitate comparisons, the plots for the comparison between normals and SCKC y IOP-CCT matched normals and SCKC have been grouped.

New comment: Page 8, Line 285. In funding section,  we add particular help that has contributed to this article. Funding: Cátedra FISABIO-Alcon-Universidad de Valencia.

Round 2

Reviewer 1 Report

Comments solved

Author Response

Thank you very much for your approval.

Reviewer 2 Report

Overall, authors have revised their manuscript adequately, but there is still some concerns regarding following issue.

Regarding definition of subclinical keratoconus:

In line 97-100: authors stated that diagnosis of SCKC was made when their topography was normal and no focal or inf. steepening pattern. However, what previous studies regarding SCKC used diagnosis criteria as following: "videokeratographic and tomographic pattern of localized steepening in the posterior or anterior corneal surface or paracentral corneal thinning, but
no clinical (keratometric, retinoscopic, or biomicroscopic) signs of keratoconus, and with a best-corrected visual acuity of 20/20 or better."

Arbelaez et al. Ophthalmology 2012;119:2231–2238.

Please find and cite a reference of SCKC definition which authors have used in the manuscript.

Author Response

Thank you for all your comments. Please find attached file for the answer.
